# In Silico Comparative Analysis of Ivermectin and Nirmatrelvir Inhibitors Interacting with the SARS-CoV-2 Main Protease

**DOI:** 10.3390/biom14070755

**Published:** 2024-06-25

**Authors:** Yuri Alves de Oliveira Só, Katyanna Sales Bezerra, Ricardo Gargano, Fabio L. L. Mendonça, Janeusa Trindade Souto, Umberto L. Fulco, Marcelo Lopes Pereira Junior, Luiz Antônio Ribeiro Junior

**Affiliations:** 1Institute of Physics, University of Brasília, Brasília 70910-900, Brazil; yurialvesy@gmail.com (Y.A.d.O.S.); ribeirojrfis@gmail.com (L.A.R.J.); 2Department of Biophysics and Pharmacology, Federal University of Rio Grande do Norte, Natal 59078-570, Brazil; katy.ksb@gmail.com (K.S.B.); umberto.laino@ufrn.br (U.L.F.); 3Department of Electrical Engineering, College of Technology, University of Brasília, Brasília 70910-900, Brazil; fabio.mendonca@unb.br (F.L.L.M.); marcelo.lopes@unb.br (M.L.P.J.); 4Department of Microbiology and Parasitology, Biosciences Center, Federal University of Rio Grande do Norte, Natal 59064-741, Brazil; janeusatsouto@gmail.com; 5Computational Materials Laboratory, University of Brasília, LCCMat, Brasília 70919-970, Brazil

**Keywords:** main protease (M^pro^), Ivermectin, Nirmatrelvir, SARS-CoV-2, molecular docking, molecular dynamics

## Abstract

Exploring therapeutic options is crucial in the ongoing COVID-19 pandemic caused by SARS-CoV-2. Nirmatrelvir, which is a potent inhibitor that targets the SARS-CoV-2 M^pro^, shows promise as an antiviral treatment. Additionally, Ivermectin, which is a broad-spectrum antiparasitic drug, has demonstrated effectiveness against the virus in laboratory settings. However, its clinical implications are still debated. Using computational methods, such as molecular docking and 100 ns molecular dynamics simulations, we investigated how Nirmatrelvir and Ivermectin interacted with SARS-CoV-2 M^pro(A)^. Calculations using density functional theory were instrumental in elucidating the behavior of isolated molecules, primarily by analyzing the frontier molecular orbitals. Our analysis revealed distinct binding patterns: Nirmatrelvir formed strong interactions with amino acids, like MET49, MET165, HIS41, HIS163, HIS164, PHE140, CYS145, GLU166, and ASN142, showing stable binding, with a root-mean-square deviation (RMSD) of around 2.0 Å. On the other hand, Ivermectin interacted with THR237, THR239, LEU271, LEU272, and LEU287, displaying an RMSD of 1.87 Å, indicating enduring interactions. Both ligands stabilized M^pro(A)^, with Ivermectin showing stability and persistent interactions despite forming fewer hydrogen bonds. These findings offer detailed insights into how Nirmatrelvir and Ivermectin bind to the SARS-CoV-2 main protease, providing valuable information for potential therapeutic strategies against COVID-19.

## 1. Introduction

Emerging in Wuhan, China, toward the end of 2019, a new acute respiratory illness caused by the novel coronavirus (SARS-CoV-2) was officially named COVID-19. The World Health Organization (WHO) declared it a global pandemic on 11 March 2020 [1]. This virus is known for its high transmissibility. It primarily spreads through human-to-human contact via contaminated respiratory secretions released during coughing, sneezing, or talking [2,3]. When in close proximity, aerosol particles can also transmit the infection, initiating the viral replication process in a new host [4].

Previously, smaller-scale epidemics, such as those in 2002, 2003, and 2012, were linked to the coronavirus, impacting various hosts, including bats, rodents, mammals, and birds [5,6], and by 2024, just over four years after the initial case in Wuhan, SARS-CoV-2 infections have affected nearly 704 million people, with approximately 7 million resulting in fatalities [7]. It is worth noting the well-known challenges in official reporting, which may lead to underestimating these values [8].

SARS-CoV-2 comprises three structural proteins and 16 non-structural proteins, among which the M^pro^, or 3CL^pro^, holds a central role in the virus’s replication mechanism, acting as a cysteine protease [9,10]—positioned strategically at the active site of M^pro^, a cysteine acts like molecular scissors, facilitating the cleavage of viral polyproteins. These lengthy protein chains, which are synthesized from the viral genome, are broken down into smaller, functionally independent units crucial for viral replication, assembly, and release. Given the vital importance of M^pro^, various research groups are currently actively engaged in the development of targeted inhibitors, aiming for potential therapeutic applications for COVID-19 [11]. Several promising candidates are in the advanced stages of clinical trials, raising optimism about their eventual incorporation into the antiviral arsenal against this persistent pathogen [12].

In this context, Alugubelli and colleagues explored the potential of Boceprevir, which is a known inhibitor of HCV’s NSP3, as a repurposed treatment for COVID-19 [13]. This compound exhibited promising inhibitory effects on the M^pro^ of SARS-CoV-2, featuring an α-ketoamide group as a warhead, along with specific structural elements, including a P1 β-cyclobutylalanine segment, a P2 dimethyl cyclopropyl proline motif, a P3 test-butyl glycine unit, and a P4 N-terminal start-butyl carbamide. Through modifications at all four positions, twenty M^pro^ inhibitors based on Boceprevir were synthesized [13]. One such derivative, Nirmatrelvir (PF-07321332), underwent characterization for its potency against M^pro^, both in vitro and in cellulo, using test tubes and 293T cells, respectively. The study involved analyzing the crystal structures of M^pro^ bound to ten of these inhibitors, alongside assessing the cytotoxicity and antiviral efficacy of four selected inhibitors.

By substituting the P1 site with a β-(S-2-oxopyrrolidin-3-yl)-alanil (Opal) residue and altering the warhead to an aldehyde, a high in vitro potency was achieved [13]. Additionally, the original structural components in P2, P3, and the P4 N-terminal positions of Boceprevir demonstrated superior performance compared with alternative chemical groups tested for their in vitro potency. In crystal structures, all inhibitors formed a covalent adduct, with the cysteine at the active site of M^pro^ [13]. Specifically, the Opal P1 residue, P2 dimethylcyclopropylproline, and P4 N-terminal tert-butylcarbamide established robust hydrophobic interactions with M^pro^, elucidating the high in vitro potency associated with compounds containing these structural features.

Andi and colleagues conducted a study that focused on developing novel antivirals and vaccines that target M^pro^ [14]. Their findings, which are supported by crystallographic evidence and binding assay data, reveal that three drugs approved for hepatitis C virus treatment, along with two drug-like compounds, establish covalent bonds with the catalytic residue Cys145 within the active site of M^pro^. Furthermore, molecular docking studies offered additional insights, aiding in the formulation of new antiviral inhibitors for SARS-CoV-2 by leveraging these drugs as lead compounds. It is worth mentioning that numerous other investigations into potential M^pro^ inhibitors have been documented in the literature [15,16,17,18].

The substrate binding site of M^pro^, which is a region known for its remarkable conservation across various coronavirus genera, was investigated by Li and colleagues [19]. They specifically examined Nirmatrelvir, which is an oral inhibitor developed by Pfizer that targets the M^pro^ of SARS-CoV-2, with demonstrated effectiveness against other coronaviruses as well. Through their analysis of these structures, the researchers unveiled a conserved binding site shared between coronaviruses, offering valuable insights into the mechanism underlying the inhibition of viral replication [19].

Viewing it from this angle, Nirmatrelvir specifically targets M^pro^ [20]. By inhibiting this protease at a critical stage of proteolysis before viral RNA replication, Nirmatrelvir effectively curbs the proliferation of SARS-CoV-2. It has exhibited potent antiviral activity against various human coronaviruses and notable off-target selectivity [20]. Additionally, animal experiments confirmed its favorable safety profile and demonstrated its capacity to reduce the lung viral load in a mouse model engineered to simulate SARS-CoV-2 infection [20]. Preclinical investigations found no evidence of mutagenic interactions with DNA [21]. Moreover, a phase I clinical trial involving healthy volunteers showed plasma concentrations surpassing the threshold of in vitro cellular antiviral potency [21].

Conversely, Ivermectin is a broad-spectrum antiparasitic medication that initially garnered attention due to its in vitro ability to inhibit SARS-CoV-2 replication, sparking interest in its potential as a COVID-19 treatment [22,23,24]. However, its off-label use for this purpose is cautiously advised. Available in both human and veterinary formulations, Ivermectin is widely used clinically to combat parasites, like Onchocerca volvulus (which causes river blindness) and Strongyloides stercoralis. Nonetheless, at elevated doses, it can lead to potentially severe adverse effects, such as nausea, vomiting, diarrhea, dizziness, and even fatalities. Large-scale clinical trials that investigated its efficacy against COVID-19 yielded mixed results. While some studies failed to demonstrate significant benefits in reducing the symptoms or duration, others suggest potential advantages [25,26]. Consequently, a consensus based on consistent, high-quality evidence regarding Ivermectin’s role in treating COVID-19 has yet to emerge in the literature.

As presented, although initially developed as an antiparasitic, Ivermectin gained notoriety for its supposed antiviral effects against COVID-19. However, robust scientific evidence is crucial to support its clinical use. Nirmatrelvir, which is supported by solid scientific evidence, shows antiviral promise with proven efficacy and a favorable safety profile. Several works are found in the literature involving Ivermectin and SARS-CoV-2, both experimental [27,28,29] and theoretical [30,31,32], although there are no theoretical investigative works that compared the inhibitory potential of Ivermectin against M^pro^. The choice of M^pro^ as a promising therapeutic target against SARS-CoV-2 is based on comprehensive and consistent experimental evidence due to its importance in viral replication and the potential of M^pro^ inhibitors to block disease progression [33,34,35]. In view of this, we conducted an in silico study to investigate the potential of interaction of Ivermectin with M^pro^, using Nirmatrelvir as an inhibitor reference. Employing molecular docking, we first examined how Ivermectin bound to M^pro^, specifically focusing on its interaction with protomer A, denoted as M^pro(A)^. Subsequently, we conducted molecular dynamics simulations for both Ivermectin and Nirmatrelvir bound to M^pro(A)^, facilitating a comprehensive analysis of their interaction profiles at the molecular level.

## 2. Methodology

### 2.1. Electronic Properties

We investigated the electronic properties of Ivermectin and Nirmatrelvir using density functional theory (DFT) and frontier molecular orbital (FMO) analyses within the Gaussian 16 software suite, Gaussian, Inc., Wallingford, CT, USA [36]. Molecular geometries were optimized through the semi-empirical Parameterization Method 6 (PM6) coupled with a steepest descent algorithm [37]. Following optimization, single-point energy calculations utilizing restricted DFT [38] were conducted to determine the energies of the highest occupied molecular orbital (HOMO), lowest unoccupied molecular orbital (LUMO), and their energy gap [39]. The analysis of HOMO/LUMO orbitals is presented here to provide contextual information about the electronic and structural differences between the investigated molecules. For a comprehensive DFT analysis, Becke’s three-parameter Lee–Yang–Parr (B3LYP) exchange–correlation functional [40] and the def2-TZVP (valence triple-zeta polarization) basis set [41] were employed.

### 2.2. Molecular Docking

Ligands and proteins were prepared using AutoDock Tools 1.5.6 (ADT), The Scripps Research Institute, San Diego, CA, USA [42]. Molecular docking simulations were employed to examine the non-covalent binding interactions between the M^pro(A)^ [20] (PDB: 7VH8) complex and Ivermectin. The crystallographic structure from the PDB contained the complex of M^pro^ and the inhibitor Nirmatrelvir. Autodock Vina, which is integrated into PyRx, The Scripps Research Institute, San Diego, CA, USA [43], was integrated with the genetic algorithm (GA) set as the default configuration for the docking simulations. This method employs a random walk simulation to explore the conformational space of the ligand while the protein remains in a fixed structure. The characterization of interactions between Ivermectin and M^pro^ was facilitated using ChimeraX, Resource for Biocomputing, Visualization, and Informatics, San Francisco, CA, USA [44] and Discovery Studio 2016, Dassault Systèmes, San Diego, CA, USA [45] programs. The simulation box encompassing the entire protein had dimensions of 60×40×65 Å^3^, with a center point at (−1.74, 18.92, −26.05) Å. Ligand screening was performed with an accuracy of ±2 Å for ligand positions and ±0.01 kcal/mol for binding affinities.

### 2.3. Molecular Dynamics

Nanoscale Molecular Dynamics (NAMD) 2.9 package, Beckman Institute for Advanced Science and Technology at the University of Illinois at Urbana-Champaign, Urbana, IL, USA [46] with the CHARMM36 force field [47] was employed for simulations. A time step of 1.0 fs was used with particle mesh Ewald (PME) electrostatics under NPT conditions (constant pressure and temperature). Langevin dynamics maintained a temperature of 300 K and pressure of 1 atm. The system was solvated with TIP3P water and neutralized with Na^+^ and Cl^−^ ions to reach 0.15 M NaCl, mimicking a physiological environment. Initially, the protein–ligand complex was restrained, followed by energy minimization and equilibration for 10 ns. A subsequent 100 ns production run allowed the system to evolve freely. The radius of gyration (Rg) and root-mean-square deviation (RMSD) and fluctuation (RMSF) analyses using Visual Molecular Dynamics (VMD) 1.9.1, Beckman Institute for Advanced Science and Technology at the University of Illinois at Urbana-Champaign, Urbana, IL, USA [48] were performed to evaluate the system behavior over time. Finally, re-docking was performed based on interactions derived from the initial docking poses.

## 3. Results

Initially, DFT calculations were performed to understand the behavior of the molecules when isolated, particularly concerning the calculated frontier molecular orbitals (HOMO and LUMO). Figure 1 displays, on the left side, the results for the HOMO and LUMO of the Ivermectin molecule, while, on the right side, it presents the same characteristics for Nirmatrelvir.

The figure analysis reveals favorable features for the interaction of Ivermectin with its target protein, indicating a potential enhancement of the bioactivity. The high HOMO energy of −6.08 eV, coupled with a low LUMO energy of −0.96 eV, resulted in a significant energy gap of 5.12 eV for Ivermectin. In the case of Nirmatrelvir, the HOMO energy was −6.75 eV, and the LUMO energy was −1.65 eV. Consequently, the energy gap was 5.10 eV. These wide gaps show efficient charge transfers of both Ivermectin and Nirmatrelvir, potentially facilitating stable interactions with the target protein. Moreover, the high energy gap implies a strong inhibition efficiency due to the minimum energy required for electron removal from the HOMO.

It was observed that ideal inhibitory molecules readily accept and donate electrons, exhibiting an electron-rich character associated with superior inhibition potential. Consistent with these principles, the previously discussed calculated LUMO energy for Ivermectin showed a good electron-accepting capacity, further supporting its potential as an effective inhibitor. Regarding the overall electronic characteristics of both Ivermectin and Nirmatrelvir, we found they were similar in both molecules. This trend implies that Ivermectin, like Nirmatrelvir, can potentially be used as an inhibitor ligand for M^pro(A)^, hindering the replication process of SARS-CoV-2.

In molecular docking, we prioritized ligand poses with more favorable binding affinity, as the lower ΔG free energy scores indicate, reflecting the most probable and stable binding modes [49]. The docking simulations revealed favorable binding affinities for Ivermectin, with the top-scoring pose presenting a ΔG of −9.0 kcal/mol. Table 1 represents the values of all binding affinities and the RMSD upper bounds and lower bounds.

From this perspective, Figure 2 details the ligand–M^pro(A)^ interactions for the best score pose of the molecular docking. At the top, we present Nirmatrelvir. This complex was obtained from Zhao and colleagues [20]. We observed that the molecule established interactions near the N-terminus of M^pro(A)^, encompassing methionines 49 and 165 (MET), histidine 41 (HIS), phenylalanine 140 (PHE), glycine 143 (GLY), threonine 190 (THR), cysteine 145 (CYS), glutamate 166 (GLU), glutamines 189 and 192 (GLN), and proline 168 (PRO). On the other hand, at the bottom of this figure, Ivermectin, in the reported best energy configuration, occupied a distinct pocket closer to the C-terminus of M^pro(A)^, interacting with alanine 154 (ALA), aspartic acid 197 (ASP), leucines 286 and 287 (LEU), methionine 176 (MET), and proline 168 (PRO). Remarkably, while Nirmatrelvir and Ivermectin were bound to separate pockets near the N- and C-termini, respectively, they converged toward the central region. The dominant interactions of Nirmatrelvir involved conventional H-bonds. In contrast, those of Ivermectin were dominated by alkyl and pi-alkyl interactions, showing that Nirmatrelvir interacted more strongly with M^pro(A)^.

Building upon the molecular docking results for Ivermectin and the Nirmatrelvir/M^pro(A)^ complex discussed earlier, MD simulations of these systems were conducted to explore the interaction dynamics of both molecules with M^pro(A)^ under ambient temperature and pressure. This aimed to verify their stability in the presence of external factors and their persistence as inhibitors throughout the simulation.

Figure 3 illustrates the temporal evolution of the RMSD (a), RMSF (b), and radius of gyration (Rg) (c) simultaneously for M^pro(A)^ in the Apo form (green curves), Nirmatrelvir (blue curves), and Ivermectin (red curves) over the 100 ns simulation time. The middle panel also presents the binding free energy (ΔG) (d), and the lower panel indicates the number of H-bonds for Nirmatrelvir (e) and Ivermectin (f). The average RMSD values for M^pro(A)^ in the Apo form was 2.15 Å; for M^pro(A)^ with Nirmatrelvir, it was 2.00 Å; and with Ivermectin, it was 1.87 Å. These values, hovering around 2.00 Å, suggest that the reference structures were highly similar to the initial structures, signifying that both ligands remained securely bound to the cavity throughout the simulation time. The RMSD curves exhibited notable similarity, characterized by low fluctuations, showing that Ivermectin has the potential to act as an effective ligand against M^pro(A)^. Moreover, in the interval between 25 ns and 45 ns of the simulation, our results indicate that Nirmatrelvir experienced more pronounced RMSD fluctuations than Ivermectin, followed by increased stability beyond this point. These results imply a more pronounced stability of M^pro(A)^ in complex with Ivermectin. On the other hand, RMSF analysis was employed to assess the dynamic behavior of the M^pro(A)^ protein when bound to Nirmatrelvir and Ivermectin. Both complexes exhibited minimal amino acid fluctuations throughout the 100 ns simulations when compared with M^pro(A)^ in the Apo form, as illustrated by their RMSF plots. Thus, both ligands significantly stabilized the M^pro(A)^ protein. The Rg measured for the two systems (protein with Ivermectin and protein with Nirmatrelvir) varied between 22.01 Å and 22.66 Å, respectively. This result indicates that the protein atoms were distributed in a similar way along its axis during the simulation in the presence of both ligands. The results indicate that the ΔG for the binding of Ivermectin and Nirmatrelvir with the target protein had negative average values of −8.25 kcal/mol and −8.50 kcal/mol, respectively. These values indicate that the interaction between the ligands and the protein was spontaneous and favorable, suggesting a high affinity between them. It is worth noting that the average affinity value for Ivermectin (−8.25 kcal/mol) was close to the molecular docking value of 9.0 kcal/mol, which corroborated the strength of the interaction. Nirmatrelvir, on the other hand, had a slightly higher average affinity (−8.50 kcal/mol), indicating an even more stable bond with the protein.

In terms of the H-bond numbers, Figure 3e,f shows that the M^pro(A)^ complex with Ivermectin exhibited superior stabilization over 100 ns compared with the complex with Nirmatrelvir. The temporal progression of hydrogen bond formation between the protein and ligands was also scrutinized to glean insights into the molecular interactions. The analysis uncovered that Nirmatrelvir initially formed four hydrogen bonds with the M^pro(A)^ protein, with two and three persisting and alternating throughout the simulation. In contrast, Ivermectin formed two to three initial hydrogen bonds, with only one remaining stable throughout the simulation.

From the above-discussed MD simulations, we emphasize the snapshots at 0 ns, 50 ns, and 100 ns (see Figure 4). These visualizations unveil a notable difference in conformational flexibility, wherein both ligands maintained binding to M^pro(A)^ throughout the simulations. However, Nirmatrelvir exhibited a high conformational change compared with Ivermectin. Notably, the flexibility of the loop regions in the M^pro(A)^ protein played a critical role in retaining the ligands, like Ivermectin and Nirmatrelvir. These flexible loops can adapt their shapes, forming tight interactions with the ligands and creating a more favorable binding pocket. This flexibility allows for the formation of additional hydrogen bonds and other stabilizing forces, which is crucial for strong ligand retention.

Finally, Figure 5 delineates the interaction profiles of Nirmatrelvir and Ivermectin with M^pro(A)^ at 0 ns, 50 ns, and 100 ns, which represent the molecular dynamics snapshots discussed earlier. At 0 ns, Nirmatrelvir (a) interacted with the following amino acids: methionines 49 and 165 (MET); histidines 41, 163, and 164 (HIS); glutamate 166 (GLU); serine 144 (SER); threonine 189 (THR); glutamine 192 (GLN); and leucine 141 (LEU). At the 50 ns mark, Nirmatrelvir (c) engaged with amino acids, including methionines 49 and 165 (MET); histidines 41, 163, and 164 (HIS); phenylalanine 140 (PHE); cysteine 145 (CYS); glutamate 166 (GLU); and asparagine 142 (ASN). In the subsequent 100 ns interval (e), the Nirmatrelvir/M^pro(A)^ complex maintained interactions with methionines 49 and 165 (MET), histidine 41, glutamate 166 (GLU), asparagine 142 (ASN), glutamine 192 (GLN), and leucine 167 (LEU). Conversely, Ivermectin’s interactions persisted at 0 ns (b), 50 ns (d), and 100 ns (f) with the amino acids tyrosine 237 (THR) and leucine 287 (LEU). Notably, Ivermectin exhibited a more enduring interaction pattern with the same amino acids over time, while Nirmatrelvir interacted with approximately half of the amino acids at 0 ns and 50 ns compared with 100 ns. Consistent with the docking results, Nirmatrelvir predominantly relied on conventional hydrogen bonds, whereas Ivermectin primarily engaged in alkyl and pi-alkyl interactions. Consequently, the more robust nature of hydrogen bonds contributed to Nirmatrelvir forming a higher number of interactions with M^pro(A)^.

## 4. Summary and Conclusions

In conclusion, this study aimed to investigate the distinct binding modes exhibited by Nirmatrelvir and Ivermectin with the SARS-CoV-2 M^pro(A)^ from the perspective of understanding the nature of these molecules as inhibitors in the viral replication process. In this regard, Nirmatrelvir demonstrated engagement with the M^pro(A)^, forming numerous interactions involving amino acids, such as methionines 49 and 165 (MET); histidines 41, 163, and 164 (HIS); phenylalanine 140 (PHE); cysteine 145 (CYS); glutamate 166 (GLU); and asparagine 142 (ASN). Through molecular dynamics simulations, the RMSD analysis revealed a stable binding of Nirmatrelvir, with an RMSD value of around 2.0 Å, indicating its secure attachment to the binding pocket over the 100 ns simulation period.

On the other hand, Ivermectin exhibited good stability, maintaining prolonged interactions with specific amino acids, such as tyrosines 237 and 239 (THR) and leucines 271, 272, and 287 (LEU). The RMSD for the Ivermectin-M^pro(A)^ complex was 1.87 Å, reflecting its consistent and secure binding throughout the simulation. It is worth noting that Ivermectin interacted with approximately the same set of amino acids at both 50 ns and 100 ns, demonstrating enduring interaction patterns over time.

This diversified exploration of ligand–protein interactions emphasized the intricate nature of these binding profiles. Nirmatrelvir, which relied on conventional hydrogen bonds, engaged in a significant number of stronger interactions with M^pro(A)^, contributing to its strong binding. However, Ivermectin primarily utilized alkyl and pi-alkyl interactions, indicating a different binding mode.

## Figures and Tables

**Figure 1 biomolecules-14-00755-f001:**
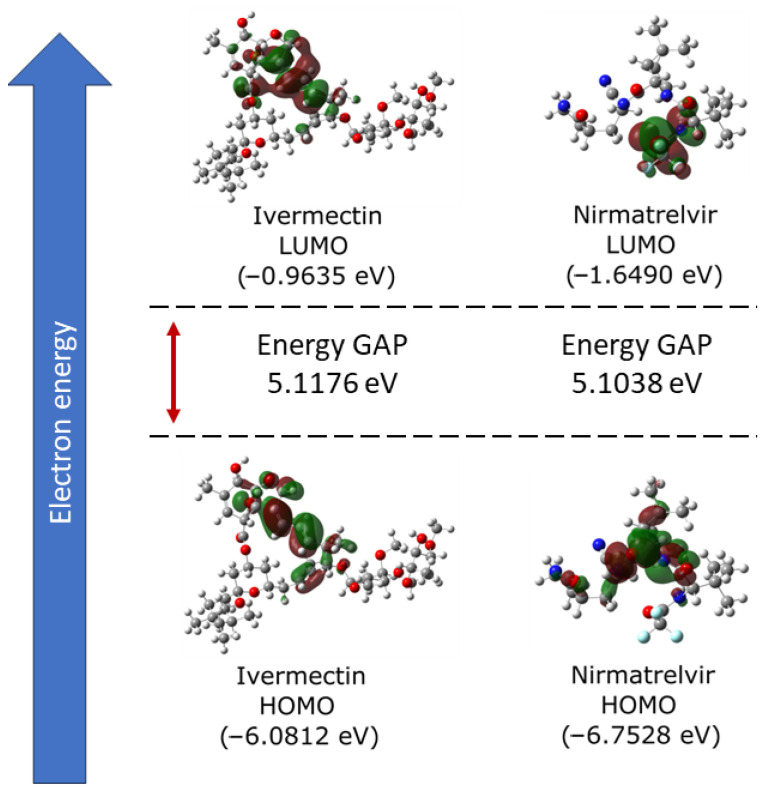
Analysis of the frontier molecular orbitals (FMOs) of Ivermectin and Nirmatrelvir, depicting their LUMOs and HOMOs, and the energy GAPs (eV) between them.

**Figure 2 biomolecules-14-00755-f002:**
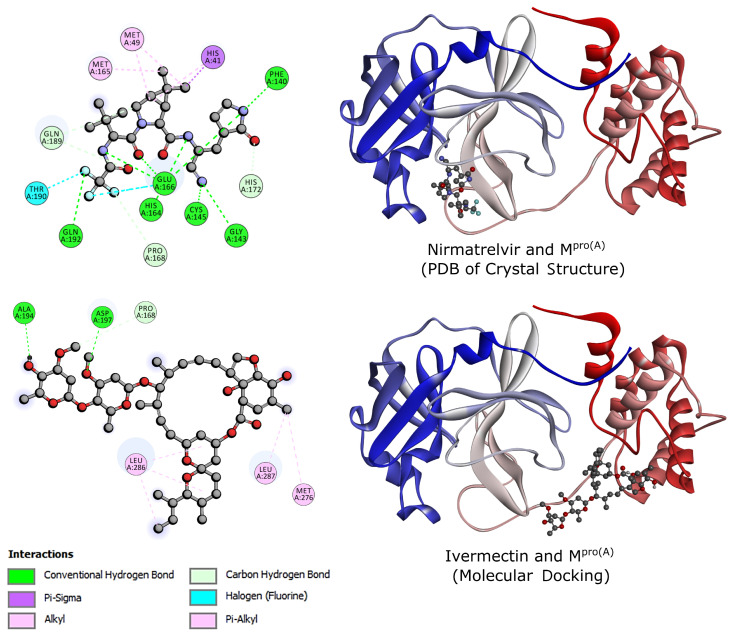
Crystal structure of Nirmatrelvir and M^pro(A)^ from RCSB PDB Protein Data Bank (**up**) and structure of Ivermectin and M^pro(A)^ from molecular docking (**down**). The blue-to-red color band represents the orientation from the N-terminus to the C-terminus of M^pro(A)^.

**Figure 3 biomolecules-14-00755-f003:**
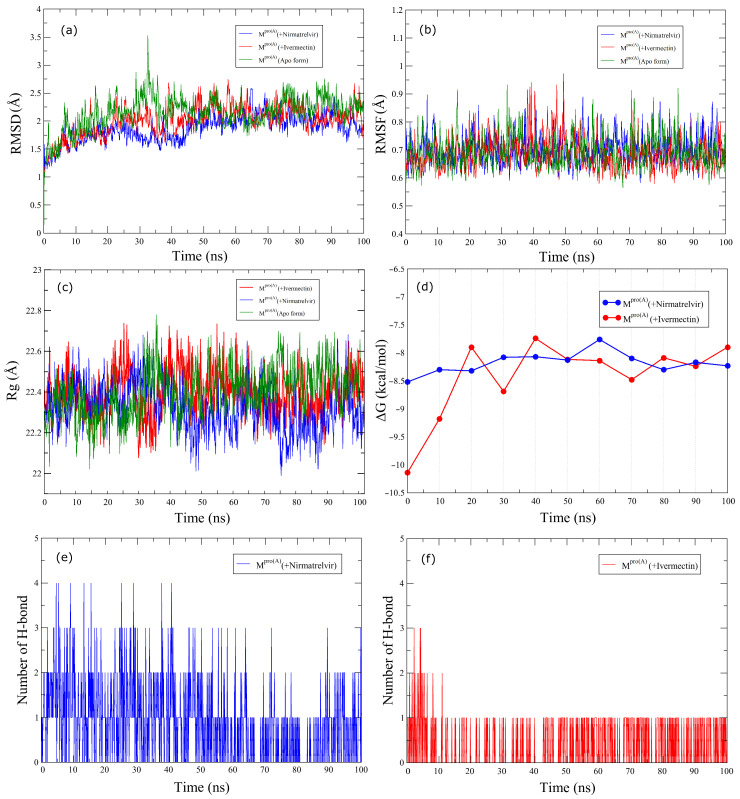
Root-mean-square deviation (**a**), root-mean-square fluctuation (**b**), radius of gyration (**c**), binding free energy (**d**), and number of hydrogen bonds (**e**,**f**) of M^pro(A)^ in the Apo form and complex with Nirmatrelvir and Ivermectin throughout the 100 ns simulation.

**Figure 4 biomolecules-14-00755-f004:**
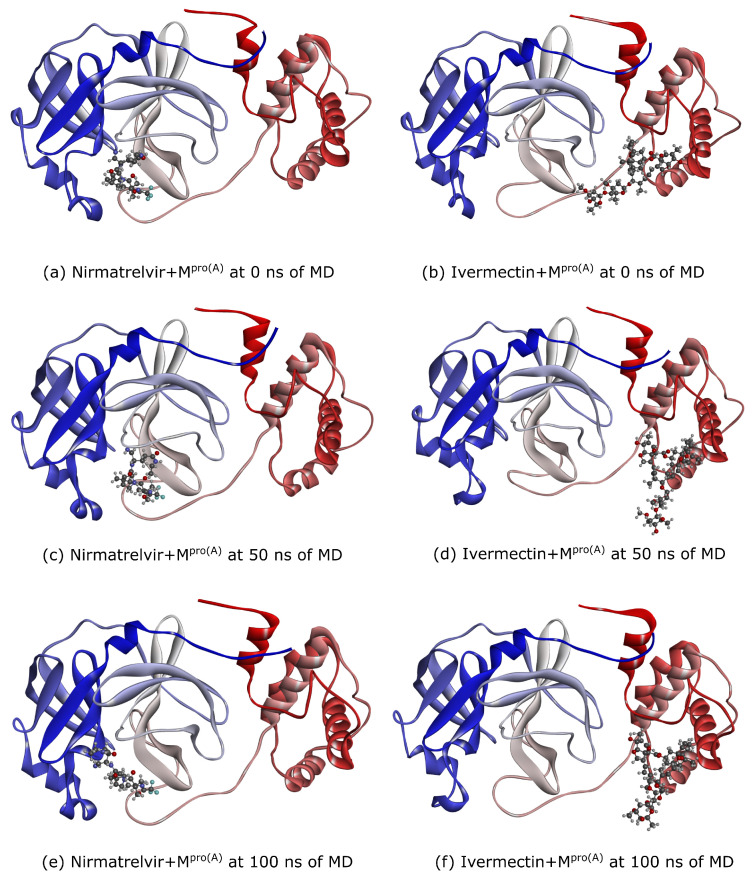
By employing the optimal docking pose (see Figure 2), we examined snapshots at 0 ns (**a**,**b**), 50 ns (**c**,**d**), and 100 ns (**e**,**f**). These depictions reveal a significant disparity in the conformational flexibility: while both ligands consistently maintained their binding to M^pro(A)^ throughout the simulations, Nirmatrelvir exhibited a markedly more pronounced conformational variation compared with Ivermectin. The red, black, cyan, blue, and grey spheres represent the oxygen, carbon, fluorine, nitrogen, and hydrogen atoms, respectively.

**Figure 5 biomolecules-14-00755-f005:**
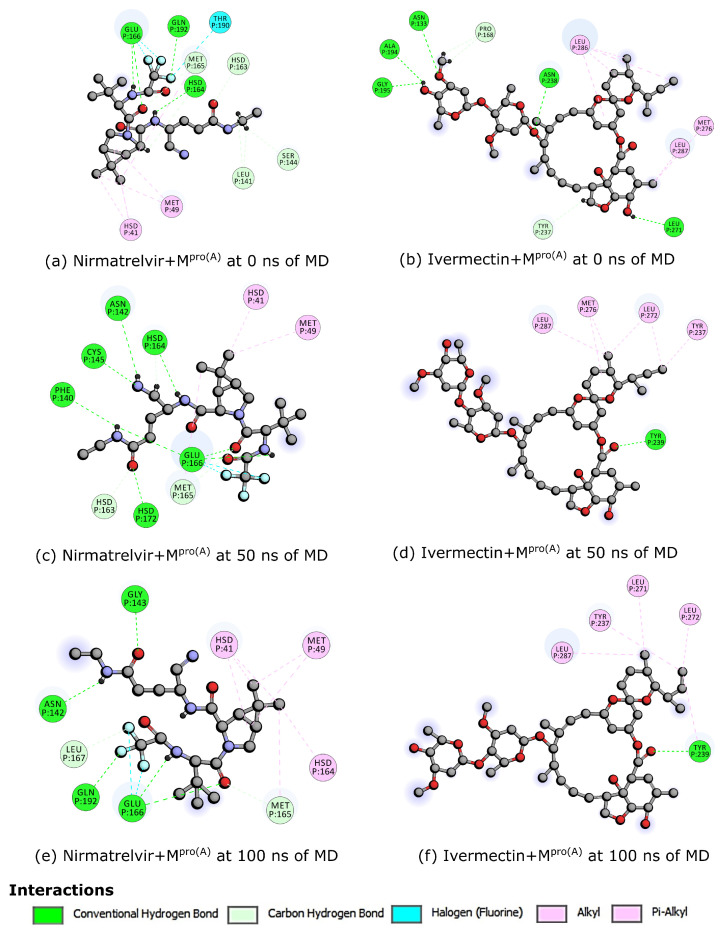
Interactions of Nirmatrelvir and Ivermectin with M^pro(A)^ at 0 ns (**a**,**b**), 50 ns (**c**,**d**), and 100 ns (**e**,**f**). The red, black, cyan, and blue spheres represent the oxygen, carbon, fluorine, and nitrogen atoms, respectively.

**Table 1 biomolecules-14-00755-t001:** AutoDock Vina binding affinity (ΔG) and RMSD upper bound/lower bound for Ivermectin/M^pro(A)^ complex.

Ranking Score	ΔG (kcal/mol)	Distance from Best Mode (Å)	
		**RMSD Upper Bound**	**RMSD Lower Bound**
1	−9.0	0.0	0.0
2	−8.7	10.871	5.250
3	−8.7	11.202	5.632
4	−8.7	15.574	8.398
5	−8.7	13.633	2.868
6	−8.4	18.253	12.631
7	−8.3	23.138	15.835
8	−8.3	23.125	15.631
9	−8.1	9.787	5.067

## Data Availability

Data are contained within the article.

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
