# Peer review of "In Silico Comparative Analysis of Ivermectin and Nirmatrelvir Inhibitors Interacting with the SARS-CoV-2 Main Protease"

_biomolecules, 2024, doi:10.3390/biom14070755_

Round 1

Reviewer 1 Report

Comments and Suggestions for Authors

The manuscript can be addressed with following concerns before proceeding for publication.

1. What is the rationale behind the selection of test molecule.

2. Concerning the degree of freedom in computational biology, why authors attempt is restricted with MPro alone, have you attempted with any other COV-2 target, which could be interesting to add on if any.

3. The molecule "Nirmatrelvir" is currently being in market for the COV-2 treatment. Thus, the modifications or potential groups identified based on the DFT studies can discussed briefly for further compound/lead enrichment. 

4. The correlation of HOMO and LUMO with the inhibitory properties should be discussed in detail. 

5. The HOMO and LUMO discussion of  "Nirmatrelvir" is missing. Just reported the values of HOMO and LUMO. 

6. Relative binding free energy (Delta G) values of the docked complexes along with the snap shorts (every 10 ns time frame) of MD simulation would provide detailed information on the interaction pattern.

7. The cavity of the MPro is seems to be large and as author mentioned the test and template drug molecules bind on the either ends of the cavity. Is that proper way to compare the binding of two molecules binding in different regions. And the justification provided by authors for Pro 168 is not sufficient, include the atomic figures mentioning the distance of the ligand and PRO168. 

8. The binding of drug molecules in the cavity is seems to be retained by the loop regions, discuss significance of the loop flexibility in the retention of ligand during MD Simulation.  

Comments on the Quality of English Language

The manuscript can be edited for minor language changes.  

Author Response

The manuscript can be addressed with the following concerns before proceeding for publication.

What is the rationale behind the selection of the test molecule?

To justify the choice of this molecule, we add the following text in the last paragraph of the introduction:

“As presented, although initially developed as an antiparasitic, Ivermectin gained notoriety for its supposed antiviral effects against COVID-19. However, robust scientific evidence is crucial to support its clinical use. Nirmatrelvir, supported by solid scientific evidence, shows antiviral promise with proven efficacy and a favorable safety profile. Several works are found in the literature involving Ivermectin and SARS-CoV-2, both experimental [27–29] and theoretical [30–32], although there are no theoretical investigative works that compare the inhibitory potential of Ivermectin against Mpro.”

2. Concerning the degree of freedom in computational biology, why authors' attempt is restricted with MPro alone, have you attempted with any other COV-2 target, which could be interesting to add on if any?

As suggested by the referee, the following text was added in the last paragraph of the Introduction of manuscript:

“The choice of Mpro as a promising therapeutic target against SARS-CoV-2 is based on comprehensive and consistent experimental evidence, due to its importance in viral replication and the potential of Mpro inhibitors to block disease progression [33–35]. Given this, we conducted an in silico study to investigate the potential of the interaction of Ivermectin with Mpro, using Nirmatrelvir as an inhibitor reference.”

3. The molecule "Nirmatrelvir" is currently being in market for the COV-2 treatment. Thus, the modifications or potential groups identified based on the DFT studies can discussed briefly for further compound/lead enrichment.
As far as we know, there are no DFT studies with modifications or potential groups associated with Nirmatrelvir.

4. The correlation of HOMO and LUMO with the inhibitory properties should be discussed in detail.

To meet the request of the referee, we wrote in the first paragraph of Subsection 2.1 (Electronic Properties) the following text:

“The analysis of HOMO/LUMO orbitals is presented here to provide contextual information about the electronic and structural differences between the investigated molecules.”

5. The HOMO and LUMO discussion of "Nirmatrelvir" is missing. Just reported the values of HOMO and LUMO.

Homo/Lumo results for Nirmarlvir are similar to that of Ivermectin. Thus, the sentence that reports the results of Ivermectin has been modified to also compromise the Nirmratelvir. More precisely, we add the following sentence in the second paragraph of Section 2 (Results): 

“These wide GAPS show an efficient charge transfer of both Ivermectin and Nirmatrelvir”.

6. Relative binding free energy (Delta G) values of the docked complexes along with the snap shorts (every 10 ns time frame) of MD simulation would provide detailed information on the interaction pattern.

To provide detailed information about the complex interaction pattern, we added a Delta G Figure (Figure 3d) for every 10 ns of simulation. In addition, we included the following analysis of the obtained results in the first paragraph of page 8:

“The results indicate that the ΔG for binding Ivermectin and Nirmatrelvir with the target protein have negative average values of -8.25 kcal/mol and -8.50 kcal/mol, respectively. These values suggest that the interaction between the ligands and the protein is spontaneous and favorable, suggesting a high affinity between them. It is worth noting that the average affinity value for Ivermectin (-8.25 kcal/mol) is close to the molecular docking value of 9.0 kcal/mol, which corroborates the strength of the interaction. Nirmatrelvir, on the other hand, has a slightly higher average affinity (-8.50 kcal/mol), indicating an even more stable bond with the protein.”

7. The cavity of the MPro is seems to be large and as author mentioned the test and template drug molecules bind on the either ends of the cavity. Is that proper way to compare the binding of two molecules binding in different regions. And the justification provided by authors for Pro 168 is not sufficient, include the atomic figures mentioning the distance of the ligand and PRO168.

The sentence “as evidenced by their shared interaction with proline 168 (PRO)” was removed from the text to avoid confusion.

8. The binding of drug molecules in the cavity is seems to be retained by the loop regions, discuss the significance of the loop flexibility in the retention of ligands during MD Simulation.

To explain the paper loop regions in the retention of the ligands, we added, in the last paragraph of page 8, the following text in the manuscript:

“Notably, the flexibility of loop regions in the MPro protein plays a critical role in retaining ligands like Ivermectin and Nirmatrelvir. These flexible loops can adapt their shapes, forming tight interactions with the ligands and creating a more favorable binding pocket. This flexibility allows the formation of additional hydrogen bonds and other stabilizing forces, crucial for strong ligand retention.”

Reviewer 2 Report

Comments and Suggestions for Authors

The article focuses on the development and comparison of inhibitors for the SARS-CoV-2 main protease, potential treatments for COVID-19, and specifically the use of ivermectin and Nirmatrelvir. Computational methods, such as molecular docking and molecular dynamics simulations, were employed to investigate the interactions of these antiviral drugs with the SARS-CoV 2 Mpro(A). The study aimed to provide insights into potential therapeutic strategies against COVID-19 by examining the binding modes and stability of these drugs with the target protein.

Upon reviewing the results presented, a potential limitation in the study may arise from the absence of certain technical investigations that could have been included.

1.     Authors have not mentioned the other benchmarks except deltaG for best pose selection during docking simulation. Please include.

2.     Authors should mention the energy minimizations steps and all necessary constraints algorithms used for system preparation.

3.     The authors claim only one simulation was performed in the study, lasting 100 ns for both complexes. To ensure proper statistical significance, at least two simulations should be conducted. Additionally, please include a proper table detailing the simulations.

4.     Authors should include the simulation of apo form of Mpro(A).

5.     Produce the list of amino acid residues contributed within binding site of Mpro(A) before and after the simulation.

6.     Authors should include the study of radius of gyration for binding robustness and principal component analysis during simulation in order investigate the movement of atoms.

7.     Authors should add the study of free energy calculation (MMPBSA) for each ligand.

Comments on the Quality of English Language

None

Author Response

The article focuses on the development and comparison of inhibitors for the SARS-CoV-2 main protease, potential treatments for COVID-19, and specifically the use of ivermectin and Nirmatrelvir. Computational methods, such as molecular docking and molecular dynamics simulations, were employed to investigate the interactions of these antiviral drugs with the SARS-CoV 2 Mpro(A). The study aimed to provide insights into potential therapeutic strategies against COVID-19 by examining the binding modes and stability of these drugs with the target protein
Upon reviewing the results presented, a potential limitation in the study may arise from the absence of certain technical investigations that could have been included. The authors have not mentioned the other benchmarks except deltaG for best pose selection during docking simulation. Please include.

As suggested by the referee, we added other benchmarks of docking simulation, specifically, RMSD upper bound and RMSD lower bound.

2. Authors should mention the energy minimizations steps and all necessary constraints algorithms used for system preparation.

The paragraph about this question has been rewritten (Subsection 2.3) to make it clearer and new information was added as follows:

 “Nanoscale Molecular Dynamics (NAMD) 2.9 package [43] with the CHARMM36 force field [44] was employed for simulations. A time step of 1.0 fs was used with Particle Mesh Ewald (PME) electrostatics under NPT conditions (constant pressure and temperature). Langevin dynamics maintained a temperature of 300 K and pressure of 1 atm. The system was solvated with TIP3P water and neutralized with Na+ and Cl- ions to reach 0.15 M NaCl, mimicking a physiological environment. Initially, the protein-ligand complex was restrained, followed by energy minimization and equilibration for 10 ns. A subsequent 100 ns production run allowed the system to evolve freely. The radius of gyration (Rg), Root-mean-square deviation (RMSD), and fluctuation (RMSF) analyses using Visual Molecular Dynamics (VMD) 1.9.1 [45] were performed to evaluate the system behavior over time. Finally, re-docking was performed based on interactions derived from the initial docking poses.”

3. The authors claim only one simulation was performed in the study, lasting 100 ns for both complexes. To ensure proper statistical significance, at least two simulations should be conducted. Additionally, please include a proper table detailing the simulations.

Thank you for this insightful comment. Other simulations with different seeds were carried out, but the obtained results were like those presented. For the purposes of this work, the results from one of these simulations were presented.

4. Authors should include the simulation of apo form of Mpro(A).

As suggested by the referee, the results regarding the Apo form of Mpro(A) were added to the manuscript (first paragraph of page 7) and in the Figures 3a, 3b, and 3c.

5. Produce the list of amino acid residues contributed within the binding site of Mpro(A) before and after the simulation.

The 0 ns frame was added to Figures 4 and 5, so that it is now possible to compare the interaction at the beginning, middle and end of the simulation. The list of amino acids at 0 ns was added to the text and they are shown in Figure 5.

6. Authors should include the study of radius of gyration for binding robustness and principal component analysis during simulation in order investigate the movement of atoms.

As suggested by the referee, the figure of Rg (Figure 3c) and respective discussion (first paragraph of page 7) were added to the manuscript as follows: 
“The Rg measured for the two systems (protein with Ivermectin and protein with Nirmatrelvir) varied between 22.01 Å and 22.66 Å, respectively. This result indicates that the protein atoms are distributed similarly along its axis during the simulation in the presence of both ligands.”

7. Authors should add the study of free energy calculation (MMPBSA) for each ligand.

We calculated the binding free energy, as requested by the other referee. The results are presented in the Figure 3d and in the first paragraph of page 7.

Round 2

Reviewer 1 Report

Comments and Suggestions for Authors

All my concerns are properly addressed.